# The Psychological Impact of Coronavirus Disease 2019 on Patients Attending a Tertiary Healthcare Facility in Pakistan: A Cross-Sectional Study

**DOI:** 10.3390/healthcare10061049

**Published:** 2022-06-05

**Authors:** Umar Abdul Ali, Nadia Iftikhar, Hafsa Amat-ur-Rasool, Mehboob Ahmed, Javeria Hafeez, Wayne G. Carter

**Affiliations:** 1Pak-Emirates Military Hospital, Rawalpindi 46000, Pakistan; umar_siddique_786@yahoo.com (U.A.A.); nadiaiftikhar12@gmail.com (N.I.); 2Institute of Microbiology and Molecular Genetics, University of the Punjab, Lahore 54590, Pakistan; hafsa.phd@gmail.com (H.A.-u.-R.); mehboob.mmg@pu.edu.pk (M.A.); 3School of Medicine, University of Nottingham, Royal Derby Hospital Centre, Uttoxeter Road, Derby DE22 3DT, UK; 4Combined Military Hospital, Rawalpindi 46000, Pakistan; javerianauman@gmail.com

**Keywords:** anxiety, COVID-19 psychological impact, depression, PTSD, stress

## Abstract

The COVID-19 pandemic has had a damaging impact on global health. Post-infection, patients may experience mental health difficulties and therefore require suitable psychological treatment and support. The objective of this study was to identify the psychological impact of COVID-19 on patients who were recovering from the physical effects of the disease, and to examine socio-demographic correlates within one month of treatment at a tertiary healthcare facility in Pakistan. A cross-sectional study was employed that utilized the Depression Anxiety Stress Scale-21 and Post-Traumatic Stress Disorder (PTSD) Checklist for DSM-5. A questionnaire was administered to 250 patients, with data collected over three months. Mild to extremely severe scores of depression, anxiety and stress were reported by approximately 43%, 52% and 42% of participants, respectively, and 8% developed PTSD. The incidence of depression, anxiety, stress or PTSD was not significantly associated with gender, age or previous interaction with COVID-19 patients. Depression was significantly associated with levels of education, severity of COVID-19 disease and a patient’s current condition. Anxiety was associated with healthcare worker status. The severity of disease and a patient’s current condition were also linked to the levels of anxiety, stress and the presence of PTSD. Collectively, these results indicate that a high percentage of patients recovering from COVID-19 experience psychological distress.

## 1. Introduction

The coronavirus disease 2019 (COVID-19) is a global pandemic that started in Wuhan, the capital city of Hubei province, China, in December 2019, with an initial mortality of approximately 3% [1]. Cases of COVID-19 soon emerged in other countries including Pakistan, which recorded its first case in Karachi, the capital city of Sindh, on 26 February 2020, and then cases rapidly spread to other Pakistani regions [2].

COVID-19 is caused by a coronavirus also known as the severe acute respiratory syndrome coronavirus-2 (SARS-CoV-2) that causes respiratory disease and is transmitted by inhalation of virus particles due to contact with infected individuals [3,4]. The disease symptoms appear in 2–14 days after first contact, with symptoms that include fever, dry cough, difficulty breathing, fatigue and loss of taste and/or smell. The effects on an individual range from asymptomatic, to mild, to acute respiratory distress, pneumonia, and multiple-organ failure. The disease affects more severely the elderly population and patients with underlying health conditions and has a mortality of 2–3% in unvaccinated individuals [3,4].

A major concern with COVID-19 is its rapid transmission and the associated increase in the number of affected individuals and subsequent daily deaths worldwide. For those that survive and their families, and for the healthcare workers that have provided care, there may be psychological impacts [5]. The consequences of the COVID-19 pandemic have impacted the mental health of the public, including increased stress, anxiety, depression, hopelessness, and uncertainty [6]. Different psychometric scales and measures such as the Depression, Anxiety and Stress Scale-21 Items (DASS-21), Impact of Events Scale-Revised (IES-R) and Post-Traumatic Stress Disorder Checklist for DSM-5 (PCL-5) have been used to assess the psychological effects of COVID-19 on individuals [7,8]. These measures are calculated with the help of standardized questionnaires and scoring methods [6].

The psychological impact on the general population of different communities, including those of Pakistan, has been due to lockdown events, fear of getting infected and loneliness due to quarantine [9]. Healthcare workers themselves have also reported psychological distress [10]. However, more studies are required that specifically focus on the post-traumatic psychological effects of COVID-19 and associated psychopathological measures [11]. Previous studies have reported some chronic psychological effects such as fatigue, restlessness, depression and post-traumatic stress disorder (PTSD) for patients that had survived the Severe Acute Respiratory Syndrome (SARS) and Middle East Respiratory Syndrome (MERS) outbreaks [12,13]. Similarly, there is a need to assess the mental health of patients that have been treated for COVID-19 [14], in order to put in place appropriate healthcare support. Indeed, mental health conditions are one of the leading causes of disability and patients with severe mental health conditions die prematurely [15].

Therefore, the objective of this study was to identify the potential psychological impact of COVID-19 on patients who were recovering from the physical effects of the disease, and to consider socio-demographic correlates, after treatment at a tertiary healthcare facility in Pakistan.

## 2. Materials and Methods

### 2.1. Study Design and Setting

A cross-sectional study was employed to assess the psychological impacts on treated patients of COVID-19 within one-month post-treatment. A questionnaire was formulated both online as well as on paper (Appendix A). The questionnaire was also translated into Urdu, the national language of Pakistan, for easy understanding and interpretation by the local population. The questionnaire was administered cross-sectionally and data were collected over three months (21 July–21 October 2020) from patients attending follow-up at a tertiary healthcare facility in Rawalpindi, Pakistan: one of the largest healthcare facilities dealing with COVID-19 patients in Pakistan. Ethical approval was obtained from the Institutional Ethical Committee of Pak-Emirates Military Hospital, Rawalpindi, Pakistan, bearing number A/28/BC/165/2020, dated 17 July 2020.

### 2.2. Participants and Sample Size

A total of 252 patients were recruited for the survey using convenience sampling; however, two patients were excluded due to a prior history of psychiatric illness. Inclusion criteria were patients treated for COVID-19 within the last month, patients that could understand and respond to the questionnaire and those that had a medical record available detailing the severity of their COVID-19 illness. Patients with a previous history of psychiatric illness were excluded from the study. Data were completed over three months. The objectives of the study were explained to all the study participants and informed consent was obtained. Patients were asked to answer a structured questionnaire so that interviewer bias could be eliminated. The questionnaire was administered at a single time point and seven participants were excluded because they did not complete the questionnaire.

### 2.3. Questionnaire Development

The questionnaire developed for the study comprised of three parts: (a) demographic data; (b) assessment of depression, anxiety and stress using DASS-21; and (c) assessment of post-traumatic stress disorder using the PTSD Checklist for DSM-5 (PCL-5).

#### 2.3.1. Demographic Data

The demographic data included name (optional), age, gender, education, occupation, history of interaction with COVID-19 patients and previous history of psychological problems (refer to the study questionnaire, Appendix A). Further information with regards to the severity of the disease and the current condition of the patient was also collected from the medical records and follow-up proformas by the study interviewers.

#### 2.3.2. Severity of COVID-19 

The severity of COVID-19 was classified as mild if the patient was symptomatic but displayed no findings on the chest radiography nor showed any decrease in oxygen saturation less than 93% on room air. Disease severity was classified as moderate if symptomatic patients displayed changes on the chest radiography but no drop in oxygen saturation less than 93% on room air. Disease classification was severe if symptomatic patients displayed findings on the chest radiography and had a drop in oxygen saturation requiring supplemental oxygen.

#### 2.3.3. Depression, Anxiety and Stress Scale-21 (DASS-21)

The mental health of participants was measured using the Depression, Anxiety and Stress Scale-21 items (DASS-21) [16]. The DASS-21 has also been used in previous studies related to COVID-19 patients [9,17,18,19]. Each of the three scales contains seven items, divided into subscales. Items 3, 5, 10, 13, 16, 17 and 21 comprise the Depression subscale; items 2, 4, 7, 9, 15, 19 and 20 comprise the Anxiety subscale; and items 1, 6, 8, 11, 12, 14 and 18 comprise the Stress subscale. All subscales are rated on a four-point Likert scale [20] ranging from 0 (never) to 3 (almost always). DASS-21 outcome scores are classified into five ranges: normal, mild, moderate, severe, and extremely severe. For stress, a normal score is from 0–10, whereas extremely severe is 35–42. For anxiety, a normal range is 0–6, while extremely severe is 20–42. For depression, a normal score is 0–9, whereas for extremely severe cases the score is 28–48.

#### 2.3.4. Assessment of Post-Traumatic Stress Disorder using PTSD Checklist for DSM-5 (PCL-5)

Post-traumatic stress disorder (PTSD) was assessed using the PTSD Checklist for DSM-5 (PCL-5). The PTSD Checklist for DSM-5 is a 20-item self-report measure that assesses the presence and severity of PTSD symptoms [21]. Items on the PCL-5 correlate with the DSM-5 criteria for PTSD [20]. The respondents were asked how they felt about each of the 20 items over the past one month on a 5-point Likert scale ranging from 0-4, where 0 = not at all, 1 = a little bit, 2 = moderately, 3 = quite a bit and 4 = extremely. Items were summated to provide a total severity score (range = 0–80). The use of PCL-5 has recently been validated in Bangladesh [22] and used to assess PTSD in COVID-19 survivors [23].

### 2.4. Statistical Analysis

The data were analyzed using IBM SPSS Statistics version 22.0. Descriptive statistics such as frequency distribution for the data were performed. The association between different socio-demographic factors and depression, stress, anxiety and PTSD was determined using the Chi-squared test for a linear trend. The significance level was set at a *p*-value of <0.05.

## 3. Results

### 3.1. Demographics of the Study Participants

Out of a total 250 recruited patients, 243 completed the questionnaire (Figure 1), of which 53 (21.8%) participants were females and 190 (78.2%) were males. Most of the participants belonged to the 20–39 years age group (49.8%), followed by the 40–59 years (35.8%) grouping and then above 59 years (13.2%). Only three participants were below 20 years. The majority of the participants were considered highly educated, i.e., 34.2% were graduates, 21% were post-graduates and 11.1% were qualified doctors, leaving 33.7% with an education level below that of a university graduate. Overall, 28.8% were public servants, 23.0% were healthcare workers (professionals and assistants), 20.6% were pensioners, 9.5% were homemakers and 21.8% were others (including students, clerical staff, farmers, shopkeepers, etc.). Most of the participants (81.5%) did not report any prior history of interaction with COVID-19 patients. The majority of participants experienced mild to moderate COVID-19 disease severity (86.9%), and most of the patients were in a phase of recovery (75.7%). 

### 3.2. Depression, Anxiety, Stress and PTSD Measurements

The mental health of participants was measured using DASS-21, a set of three scales designed to measure depression, anxiety and stress in individuals. According to the present study, 104 (42.8%) of the participants showed signs of depression (Table 1), among which 13.5% (14 of 104) and 7.7% (8 of 104) were of the severe and extremely severe categories, respectively (Table 2). Depression was not significantly associated with gender, age or among healthcare and non-healthcare workers or those with a history of interaction with COVID-19 patients (*p* > 0.05) (Table 2). However, the level of depression was significantly different between patients of different education levels (*p* = 0.04) and was related to the severity of the disease (*p* = 0.03) as well as the patient’s current condition (*p* = 0.04) (Table 2).

An analysis of the levels of anxiety revealed that 126 (52%) of the participants showed anxiety symptoms, among which 23.8% (30 of 126) and 22.2% (28 of 126) were within the severe and extremely severe categories, respectively (Table 1). Anxiety was not significantly associated with gender, age, education level or history of interaction with COVID-19 patients (*p* > 0.05) (Table 1). The level of anxiety was linked with healthcare worker status and the severity of the COVID-19 disease, but this did not reach significance (*p* > 0.05). However, the level of anxiety was significantly higher in recovering patients (*p* = 0.03) and was most prevalent in patients with moderate COVID-19 disease (Table 1).

According to the DASS-21 Scale, 102 (42%) of the participants evidenced stress (Table 3), among which 6.9% (7 of 102) had severe and 5.9% (6 of 102) had extremely severe stress levels (Table 3). Stress was not significantly associated with gender, age, education or history of interaction with COVID-19 patients (*p* > 0.05) (Table 3). The level of stress was linked with healthcare worker status, but this did not reach significance (*p* > 0.05). However, stress levels were significantly associated with the severity of COVID-19 disease (*p* = 0.04) and were significantly higher in recovering patients (*p* = 0.01).

An assessment of PTSD among the participants was undertaken using the PTSD Checklist 5 (PCL-5) to evaluate patient symptoms within the previous month. In total, 19 (7.8%) out of 243 showed symptoms of PTSD (Table 4). PTSD was not significantly associated with gender, age, education, healthcare worker status or history of interaction with COVID-19 patients (*p* > 0.05) (Table 4). PTSD was, however, significantly associated with the severity of the disease (*p* = 0.02) and the current condition of the patient (*p* = 0.01) (Table 4).

## 4. Discussion

COVID-19 is a devastating pandemic that has caused severe physical as well as a psychological impacts on those infected [24,25]. The present study considered a broad age range of participants (15–83 years of age, with a mean age of 42 years) and was conducted to identity the presence of depression, anxiety, stress and PTSD in patients recovering from COVID-19 attending a tertiary healthcare facility in Pakistan, and to examine socio-demographic correlates. Our study found that, within one-month post-treatment, the prevalence of depression, anxiety, stress and PTSD among 243 patients of COVID-19 was approximately 43%, 52%, 42% and 8%, respectively (Table 1, Table 2, Table 3 and Table 4).

The impact of COVID-19 on the psychological health of patients has been evident across age groups. For example, in a broad 4700 participant study in Istanbul (under 30 to over 60 age group), 64% of the participants were identified as psychologically fatigued [26]. Likewise, from a study of 1143 participants (parents of children aged 3–18 years), 86% of the parents perceived the psychological effects of the COVID-19 quarantine among Italian and Spanish youth [27]. There were clinically high levels of distress in 555 adults (age range 18–76, with a mean age of 39 years) in the UK, with 68% reporting that they were very or somewhat worried about COVID-19 [28].

Our report on the percentage of COVID-19 patients that suffered depression (42.8%), when assessed in the short-term, is similar in percentage to a published study from Shenzhen, China that reported 38.1% (of 126 patients) had experienced clinically significant depression [29]. Anxiety levels in this study were less (22.2%), but the numbers of patients with stress levels were also comparable (31.0%) [29], highlighting that detection of the potential impact on the mental health of patients is evident in the short-term and is feasible to measure and quantitate, even with a relatively small sample size. Other studies also highlighted that a patient of female gender was more likely to experience a psychological impact than males [6,29], but in the cohort detailed herein, a significant difference between genders was not present, although our study had a disproportionately higher number of male participants than the other studies.

In the present study, the levels of depression were further sub-classified into mild to extremely severe categories, and severe or extremely severe cases constituting approximately 9% of the patients. A significant association between depression and level of education was detected, and this may have reflected a greater knowledge and understanding of the disease in physicians and those at the post-graduate level, although lower educational attainment can itself be predictive of depression and anxiety [30]. Depression was also significantly associated with the severity of COVID-19 disease and a patient’s current condition.

Our study reports that just over half of the study participants experienced anxiety, and this was further divided into mild to extremely severe categories, of which approximately 12% were of severe or extremely severe categories. The prevalence of anxiety was higher in non-healthcare workers, but this did not reach significance in our study. Anxiety has been reported to be higher (20.7%) among non-medical healthcare workers when compared with medical personnel (10.8%) in another COVID-19 study [31], which may also be related to improved knowledge and understanding of the disease. Anxiety was also significantly associated with the severity of the disease and a patient’s current condition.

There were 42% of the study participants with a level of stress that ranged from mild to extremely severe forms, with the majority categorized as mild. Stress was also higher in non-healthcare workers but did not reach significance. However, significant levels of stress were associated with the severity of the disease and the current condition of the patient.

Our study reports that approximately 8% of the participants developed PTSD and that this was significantly associated with the severity of the disease and the current condition of the patient.

There are some limitations to our study data since this was gathered following the first COVID-19 outbreak in Pakistan when the fear of disease was prevalent, in part due to the unavailability of vaccination. The study was also undertaken using convenience sampling and resulted in an uneven distribution between genders, with the majority of data generated from male subjects. In addition, the study number of 250 arose based on the manageability of interview numbers; therefore, external validity and generalizability of the study results to the population at large are not feasible. Nevertheless, the benefits of evaluation of the psychological impact of the disease and associated patient management are likely to be required and are still relevant problems, particularly in cohorts of patients that do not have access to established primary care and vaccination. Hence, the present study suggests that it will be valuable to consider appropriate psychological assessment and treatment options for patients of COVID-19, particularly if they display early symptoms associated with the psychological impact of the disease. Furthermore, our study results suggest that there may be some predictors of the need for psychological care and support, including an assessment of the level of education of the patient and presumed knowledge of the disease that could arise from working within a healthcare setting. These parameters underscore the importance of providing the relevant and correct information about disease transmission and risks of vaccination, as misinformation and conspiracy theories, particularly from digital sources, have helped propagate an environment of unease, anxiety, and feelings of depression [32,33].

## 5. Summary and Conclusions

In summary, our study suggests that the rapidly spreading COVID-19 pandemic has impacted the mental health of patients in recovery, such that a relatively high percentage of patients displayed signs of depression, anxiety, stress, and PTSD; these numbers were several times higher than those experienced by the general public in response to COVID-19 [34]. Collectively, the number of patients experiencing the psychological impact of COVID-19 suggests an immediate need for specialized medical care and patient management by a multidisciplinary team, such as a physician, psychologist and psychiatrist, as well as support from social rehabilitation services. Furthermore, the potential for long-lasting effects including PTSD warrants the need for ongoing psychological support which may require physical, pharmacological and psychological interventions [35]. Current medical practices may also need to be adapted to facilitate healthcare delivery peripherally, such as via telemedicine approaches and the establishment of suitable support groups, and a potential requirement to adopt lifestyle changes that may arise from domiciliary confinement [36,37].

## Figures and Tables

**Figure 1 healthcare-10-01049-f001:**
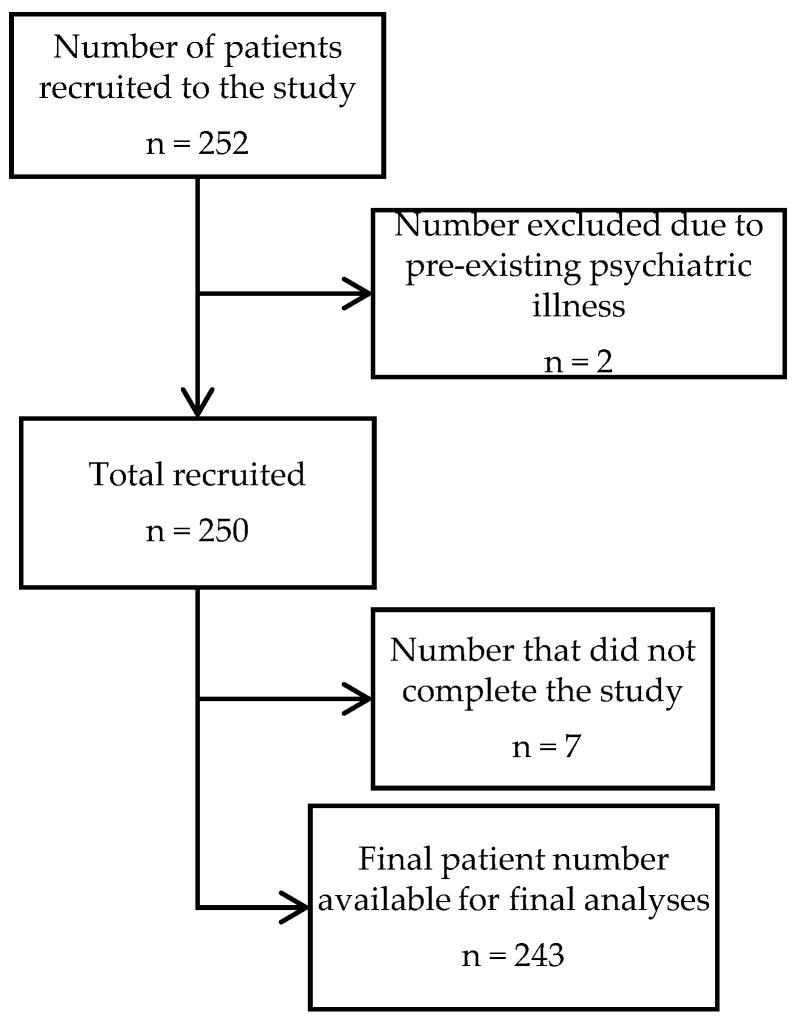
Flow diagram detailing the selection of patient participants.

**Table 1 healthcare-10-01049-t001:** Percentage distribution of demographics and anxiety levels.

			Anxiety Levels					
			Normal	Mild	Moderate	Severe	Extremely Severe	*p* Value
	Total	n	117	24	44	30	28	
		%	(48.10%)	(9.90%)	(18.10%)	(12.30%)	(11.50%)	
Gender	Male	n	94	16	39	22	19	0.31
		%	(38.70%)	(6.60%)	(16.00%)	(9.10%)	(7.80%)	
	Female	n	23	8	5	8	9	
		%	(9.50%)	(3.30%)	(2.10%)	(3.30%)	(3.70%)	
Age	<20	n	1	0	1	1	0	0.75
		%	(0.40%)	(0.00%)	(0.40%)	(0.40%)	(0.00%)	
	20–39	n	56	14	26	14	11	
		%	(23.00%)	(5.80%)	(10.70%)	(5.80%)	(4.50%)	
	40–59	n	42	7	12	12	14	
		%	(17.30%)	(2.90%)	(4.90%)	(4.90%)	(5.80%)	
	>60	n	18	3	5	3	3	
		%	(7.40%)	(1.20%)	(2.10%)	(1.20%)	(1.20%)	
Education	Below Graduate Level	n	41	10	15	7	9	0.471
		%	(16.90%)	(4.10%)	(6.20%)	(2.90%)	(3.70%)	
	Graduate	n	37	8	17	13	8	
		%	(15.20%)	(3.30%)	(7.00%)	(5.30%)	(3.30%)	
	Physicians	n	15	2	4	2	4	
		%	(6.20%)	(0.80%)	(1.60%)	(0.80%)	(1.60%)	
	Post-Graduate Level	n	24	4	8	8	7	
		%	(9.90%)	(1.60%)	(3.30%)	(3.30%)	(2.90%)	
Healthcare Workers	Yes	n	34	5	6	4	7	0.10
		%	(14.00%)	(2.10%)	(2.50%)	(1.60%)	(2.90%)	
	No	n	83	19	38	26	21	
		%	(34.20%)	(7.80%)	(15.60%)	(10.70%)	(8.60%)	
History of Interaction with COVID-19 Patients	Yes	n	27	4	3	4	7	0.35
		%	(11.10%)	(1.60%)	(1.20%)	(1.60%)	(2.90%)	
	No	n	90	20	41	26	21	
		%	(37.00%)	(8.20%)	(16.90%)	(10.70%)	(8.60%)	
Severity of COVID-19 Disease	Mild	n	59	11	15	15	9	0.13
		%	(24.30%)	(4.50%)	(6.20%)	(6.20%)	(3.70%)	
	Moderate	n	44	9	25	9	15	
		%	(18.10%)	(3.70%)	(10.30%)	(3.70%)	(6.20%)	
	Severe	n	14	4	4	6	4	
		%	(5.80%)	(1.60%)	(1.60%)	(2.50%)	(1.60%)	
Current Condition	Recovering	n	91	17	34	25	17	0.03
		%	(37.40%)	(7.00%)	(14.00%)	(10.30%)	(7.00%)	
	Same as Before Illness	n	20	6	7	2	3	
		%	(8.20%)	(2.50%)	(2.90%)	(0.80%)	(1.20%)	
	Getting Worse	n	6	1	3	3	8	
		%	(2.50%)	(0.40%)	(1.20%)	(1.20%)	(3.30%)	

**Table 2 healthcare-10-01049-t002:** Percentage distribution of demographics and depression levels.

			Depression Levels					
			Normal	Mild	Moderate	Severe	Extremely Severe	*p* Value
Total		n	139	42	40	14	8	
		%	(57.20%)	(17.30%)	(16.50%)	(5.80%)	(3.30%)	
Gender	Male	n	109	31	29	13	8	0.34
		%	(44.90%)	(12.80%)	(11.90%)	(5.30%)	(3.30%)	
	Female	n	30	11	11	1	0	
		%	(12.30%)	(4.50%)	(4.50%)	(0.40%)	(0.00%)	
Age	<20	n	2	0	0	1	0	0.83
		%	(0.80%)	(0.00%)	(0.00%)	(0.40%)	(0.00%)	
	20–39	n	71	21	19	6	4	
		%	(29.20%)	(8.60%)	(7.80%)	(2.50%)	(1.60%)	
	40–59	n	48	13	18	6	2	
		%	(19.80%)	(5.30%)	(7.40%)	(2.50%)	(0.80%)	
	>60	n	18	8	3	1	2	
		%	(7.40%)	(3.30%)	(1.20%)	(0.40%)	(0.80%)	
Education	Below Graduate Level	n	57	8	12	5	0	0.04
		%	(23.50%)	(3.30%)	(4.90%)	(2.10%)	(0.00%)	
	Graduate	n	40	22	14	4	3	
		%	(16.50%)	(9.10%)	(5.80%)	(1.60%)	(1.20%)	
	Physicians	n	15	4	4	1	3	
		%	(6.20%)	(1.60%)	(1.60%)	(0.40%)	(1.20%)	
	Post-Graduate Level	n	27	8	10	4	2	
		%	(11.10%)	(3.30%)	(4.10%)	(1.60%)	(0.80%)	
Healthcare Workers	Yes	n	34	9	9	1	3	0.66
		%	(14.00%)	(3.70%)	(3.70%)	(0.40%)	(1.20%)	
	No	n	105	33	31	13	5	
		%	(43.20%)	(13.60%)	(12.80%)	(5.30%)	(2.10%)	
History of Interaction with COVID-19 Patients	Yes	n	27	7	7	1	3	0.96
		%	(11.10%)	(2.90%)	(2.90%)	(0.40%)	(1.20%)	
	No	n	112	35	33	13	5	
		%	(46.10%)	(14.40%)	(13.60%)	(5.30%)	(2.10%)	
Severity of COVID-19 Disease	Mild	n	71	19	12	4	3	0.03
		%	(29.20%)	(7.80%)	(4.90%)	(1.60%)	(1.20%)	
	Moderate	n	49	19	26	5	3	
		%	(20.20%)	(7.80%)	(10.70%)	(2.10%)	(1.20%)	
	Severe	n	19	4	2	5	2	
		%	(7.80%)	(1.60%)	(0.80%)	(2.10%)	(0.80%)	
Current Condition	Recovering	n	109	35	23	10	7	0.04
		%	(44.90%)	(14.40%)	(9.50%)	(4.10%)	(2.90%)	
	Same as Before Illness	n	21	7	9	1	0	
		%	(8.60%)	(2.90%)	(3.70%)	(0.40%)	(0.00%)	
	Getting Worse	n	9	0	8	3	1	
		%	(3.70%)	(0.00%)	(3.30%)	(1.20%)	(0.40%)	

**Table 3 healthcare-10-01049-t003:** Percentage distribution of demographics and stress levels.

			Stress Levels					
			Normal	Mild	Moderate	Severe	Extremely Severe	*p* Value
Total		n	141	57	32	7	6	
		%	(58.00%)	(23.50%)	(13.20%)	(2.90%)	(2.50%)	
Gender	Male	n	110	43	25	7	5	0.50
		%	(45.30%)	(17.70%)	(10.30%)	(2.90%)	(2.10%)	
	Female	n	31	14	7	0	1	
		%	(12.80%)	(5.80%)	(2.90%)	(0.00%)	(0.40%)	
Age	<20	n	1	1	1	0	0	0.79
		%	(0.40%)	(0.40%)	(0.40%)	(0.00%)	(0.00%)	
	20–39	n	73	27	17	2	2	
		%	(30.00%)	(11.10%)	(7.00%)	(0.80%)	(0.80%)	
	40–59	n	46	22	12	5	2	
		%	(18.90%)	(9.10%)	(4.90%)	(2.10%)	(0.80%)	
	>60	n	21	7	2	0	2	
		%	(8.60%)	(2.90%)	(0.80%)	(0.00%)	(0.80%)	
Education	Below Graduate Level	n	50	18	12	1	1	0.35
		%	(20.60%)	(7.40%)	(4.90%)	(0.40%)	(0.40%)	
	Graduate	n	48	19	11	2	3	
		%	(19.80%)	(7.80%)	(4.50%)	(0.80%)	(1.20%)	
	Physicians	n	16	6	3	1	1	
		%	(6.60%)	(2.50%)	(1.20%)	(0.40%)	(0.40%)	
	Post-Graduate Level	n	27	14	6	3	1	
		%	(11.10%)	(5.80%)	(2.50%)	(1.20%)	(0.40%)	
Healthcare Workers	Yes	n	36	13	5	1	1	0.19
		%	(14.80%)	(5.30%)	(2.10%)	(0.40%)	(0.40%)	
	No	n	105	44	27	6	5	
		%	(43.20%)	(18.10%)	(11.10%)	(2.50%)	(2.10%)	
History of Interaction with COVID-19 Patients	Yes	n	28	11	4	1	1	0.42
		%	(11.50%)	(4.50%)	(1.60%)	(0.40%)	(0.40%)	
	No	n	113	46	28	6	5	
		%	(46.50%)	(18.90%)	(11.50%)	(2.50%)	(2.10%)	
Severity of COVID-19 Disease	Mild	n	70	25	11	2	1	0.04
		%	(28.80%)	(10.30%)	(4.50%)	(0.80%)	(0.40%)	
	Moderate	n	54	24	17	3	4	
		%	(22.20%)	(9.90%)	(7.00%)	(1.20%)	(1.60%)	
	Severe	n	17	8	4	2	1	
		%	(7.00%)	(3.30%)	(1.60%)	(0.80%)	(0.40%)	
Current Condition	Recovering	n	110	47	19	3	5	0.01
		%	(45.30%)	(19.30%)	(7.80%)	(1.20%)	(2.10%)	
	Same as Before Illness	n	24	6	7	0	1	
		%	(9.90%)	(2.50%)	(2.90%)	(0.00%)	(0.40%)	
	Getting Worse	n	7	4	6	4	0	
		%	(2.90%)	(1.60%)	(2.50%)	(1.60%)	(0.00%)	

**Table 4 healthcare-10-01049-t004:** Percentage distribution of demographics and post-traumatic stress disorder.

			PTSD		
			No	Yes	*p* Value
Total		n	224	19	
		%	(92.2%)	(7.8%)	
Gender	Male	n	177	13	0.28
		%	(72.8%)	(5.3%)	
	Female	n	47	6	
		%	(19.3%)	(2.5%)	
Age	<20	n	3	0	0.61
		%	(1.2%)	(0.0%)	
	20–39	n	114	7	
		%	(46.9%)	(2.9%)	
	40–59	n	78	9	
		%	(32.1%)	(3.7%)	
	>60	n	29	3	
		%	(11.9%)	(1.2%)	
Education	Below Graduate Level	n	78	4	0.54
		%	(32.1%)	(1.6%)	
	Graduate	n	76	7	
		%	(31.3%)	(2.9%)	
	Physicians	n	25	2	
		%	(10.3%)	(0.8%)	
	Post-Graduate Level	n	45	6	
		%	(18.5%)	(2.5%)	
Healthcare Workers	Yes	n	53	3	0.43
		%	(21.8%)	(1.2%)	
	No	n	171	16	
		%	(70.4%)	(6.6%)	
History of Interaction with COVID-19 Patients	Yes	n	42	3	0.75
		%	(17.3%)	(1.2%)	
	No	n	182	16	
		%	(74.9%)	(6.6%)	
Severity of COVID-19 Disease	Mild	n	106	3	0.02
		%	(43.6%)	(1.2%)	
	Moderate	n	89	13	
		%	(36.6%)	(5.3%)	
	Severe	n	29	3	
		%	(11.9%)	(1.2%)	
Current Condition	Recovering	n	172	12	0.01
		%	(70.8%)	(4.9%)	
	Same as Before Illness	n	36	2	
		%	(14.8%)	(0.8%)	
	Getting Worse	n	16	5	
		%	(6.6%)	(2.1%)	

## Data Availability

Data can be provided by the first author of the manuscript on request.

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
