# Peer review of "The Psychological Impact of Coronavirus Disease 2019 on Patients Attending a Tertiary Healthcare Facility in Pakistan: A Cross-Sectional Study"

_healthcare, 2022, doi:10.3390/healthcare10061049_

Round 1

Reviewer 1 Report

In this study, the authors sought to evaluate “the psychological impact of COVID-19 on patients who survived the physical effects of the disease […] in a tertiary healthcare facility in Pakistan. To this end, the authors performed a cross-sectional study within one month after having COVID-19. The study is interesting, although it has severe reporting deficiencies which need to be corrected to re-assess this manuscript since it is currently not reproducible and difficult to interpret due to very poor reporting.

Major comments:

1. The study objective is very poorly defined. Please re-write it and make sure to include all elements of your PEO question in the study objective. Exposures and outcomes need to be clearly identified. 2. It looks like the authors did not calculate a sample size for this study. Please explain in the manuscript why a sample size was not calculated and recognize this as a limitation of the study. 3. What was the sampling strategy? It looks like convenience sampling was applied. Please label it in the manuscript methods while also discussing all potential biases of this sampling method in the manuscript limitations. 4. Exposures and outcome(s) need to be clearly defined in the methods section according to STROBE recommendations. See the STROBE explanation and elaboration document for clarifications (https://journals.plos.org/plosmedicine/article?id=10.1371/journal.pmed.0040297). 5. Please add a flow diagram of patients in the study according to STROBE recommendations. See the STROBE document for clarifications. 6. This questionnaire had a very low percentage of participants who were woman. Please add this as a limitation of your study since the study had a very low representativity of women. 7. In the section on demographic data it is mentioned that information was obtained on the severity of the disease and the current condition of the patients, but it is not clearly explained how this information was obtained. It looks like the authors asked the patients to respond the severity of their disease. Since this was not based on clearly defined clinical criteria and is subject to severe bias, all analyses related to this variable need to be eliminated. 8. The statement “Once signed I cannot revoke consent” present in the informed consent section of the collection sheet present in the supplementary material is worrisome since it would represent a violation of good international clinical practices. 9. In the statistical analysis section, the authors mention that the descriptive data are presented as mean and frequency, but in no section of the results are mean data presented. The authors should describe the mean age and the age dispersion in the sample. The associations between variables were evaluated by the chi square test, but according to the types of variables, the correct test would be the linear trend chi square test, please correct the results. 10. The results presented must be carried out again considering the same categories of the DASS-21 expressed in the methodology, since the authors arbitrarily dichotomized a variable that was originally ordinal. 11. The authors mention in Table 3 the severity of depression, anxiety and stress obtained in the DASS-21, but nowhere in the methodology do they explain how this severity classification was obtained, please clarify. 12. Table 4 on the p values ​​is useless presented in this way, it should be eliminated, and each p value should be included next to the comparison made. 13. There are several other reporting deficiencies that can be addressed by adequately reporting this study according to STROBE recommendations. Please review the full STROBE explanation and elaboration document (https://doi.org/10.1371/journal.pmed.0040297) to report your manuscript accordingly. Please also provide the STROBE checklist for a cross-sectional study for your revised manuscript as supplementary material for peer-review only; this checklist can be easily built in https://www.goodreports.org/

Minor comments:

1. Please review the official abbreviations of COVID-19 and SARS-CoV-2 by the WHO and modify accordingly: https://www.who.int/emergencies/diseases/novel-coronavirus-2019/technical-guidance/naming-the-coronavirus-disease-(covid-2019)-and-the-virus-that-causes-it 2. Lines 75-80: This should be described in the methods section, not the introduction. 3. Lines 80-81: Please delete this phrase since it conveys a conclusion which should not be in the introduction section. 4. The specific study design has not been adequately labeled in the methods section. 5. Inclusion and exclusion criteria should not be complementary between each other. The information that is already conveyed within an inclusion criteriondoes not need to be mentioned in a complementary exclusion criteria. 6. Lines 98-100: This is part of your results, not your methods. 7. Lines 109-110: This should be described alongside selection criteria, not here. 8. I recommend including this reference in your discussion (DOI: 10.3390/nu13093213), because thecomponents assessed in that study, such as changes in lifestyle and weight, can affect emotional health, which were not evaluated in this study. 9. The authors mention that they calculated 95% confidence intervals, but they are never expressed in results. Please clarify it.

Author Response

Response to Reviewer #1:

Major comments:

  1. The study objective is very poorly defined. Please re-write it and make sure to include all elements of your PEO question in the study objective. Exposures and outcomes need to be clearly identified.

We thank Reviewer #1 for this prompt to better describe the research question and study objective.  The PEO (patient, population, and outcome) is now clearly defined in the abstract. The manuscript has also undergone an extensive rewrite to improve clarity regarding the research question that was addressed.

  1. It looks like the authors did not calculate a sample size for this study. Please explain in the manuscript why a sample size was not calculated and recognize this as a limitation of the study.

The sample size was a reflection of a manageable number of clinical interviews by the first author of the study.  We appreciate that this is a limitation of the study and therefore external validity and generalizability of the study results to the population at large are not feasible. We have discussed this further in the revised manuscript.

  1. What was the sampling strategy? It looks like convenience sampling was applied. Please label it in the manuscript methods while also discussing all potential biases of this sampling method in the manuscript limitations.

We thank Reviewer #1 for raising this point as the sampling method should have been better detailed.  Reviewer #1 is correct; patients were recruited to the study using convenience sampling.  This sampling method is biased and may not have applicability to the population at large.  We have discussed this further in the limitations of the discussion section.

  1. Exposures and outcome(s) need to be clearly defined in the methods section according to STROBE recommendations. See the STROBE explanation and elaboration document for clarifications (https://journals.plos.org/plosmedicine/article?id=10.1371/journal.pmed.0040297).

We thank Reviewer #1 for making this suggestion of ensuring that the manuscript follows the STROBE recommendations.  We have updated the manuscript accordingly.

  1. Please add a flow diagram of patients in the study according to STROBE recommendations. See the STROBE document for clarifications.

We thank Reviewer #1 for making this suggestion of ensuring that the manuscript follows the STROBE recommendations.  We have updated the manuscript accordingly but since the study was small in number of participants and only excluded patients due to a previous history of psychological problems, a flow diagram was not included.

  1. This questionnaire had a very low percentage of participants who were woman. Please add this as a limitation of your study since the study had a very low representativity of women.

We thank Reviewer #1 for raising this.  The convenience sampling resulted in a gender imbalance of patients, and this is a limitation of the study and has now been highlighted in the limitations section of the revised manuscript.

  1. In the section on demographic data it is mentioned that information was obtained on the severity of the disease and the current condition of the patients, but it is not clearly explained how this information was obtained. It looks like the authors asked the patients to respond the severity of their disease. Since this was not based on clearly defined clinical criteria and is subject to severe bias, all analyses related to this variable need to be eliminated.

We thank Reviewer #1 for raising this oversight.  The COVID-19 patients were assessed and monitored by clinicians and the severity of the disease established by clinical criteria. These details have now been added to the revised manuscript.

  1. The statement “Once signed I cannot revoke consent” present in the informed consent section of the collection sheet present in the supplementary material is worrisome since it would represent a violation of good international clinical practices.

This statement should have context and appropriate translation. The study was administered as a single time point questionnaire with the intention that patients would start and complete the questionnaire.  But this was not enforced and 7 of the 250 recruited patients did not complete the study, hence the study number of 243.  These details should have been added to the methods section and have now been added to the revised manuscript.

  1. In the statistical analysis section, the authors mention that the descriptive data are presented as mean and frequency, but in no section of the results are mean data presented. The authors should describe the mean age and the age dispersion in the sample. The associations between variables were evaluated by the chi square test, but according to the types of variables, the correct test would be the linear trend chi square test, please correct the results.

We apologize for this and appreciate the guidance from Reviewer #1.  All the data has been revised and new tables produced (Tables 1-4) using a Chi squared test for a linear trend. 

  1. The results presented must be carried out again considering the same categories of the DASS-21 expressed in the methodology, since the authors arbitrarily dichotomized a variable that was originally ordinal.

The results utilized both the DASS-21 as well as the post-traumatic stress disorder (PTSD) Checklist for DSM 5 (PCL-5).

  1. The authors mention in Table 3 the severity of depression, anxiety and stress obtained in the DASS-21, but nowhere in the methodology do they explain how this severity classification was obtained, please clarify.

As now detailed in the methods section, the patients were monitored via clinicians.

  1. Table 4 on the p values ​​is useless presented in this way, it should be eliminated, and each p value should be included next to the comparison made.

As detailed above, the data section has been completely reworked and presented.

  1. There are several other reporting deficiencies that can be addressed by adequately reporting this study according to STROBE recommendations. Please review the full STROBE explanation and elaboration document (https://doi.org/10.1371/journal.pmed.0040297) to report your manuscript accordingly. Please also provide the STROBE checklist for a cross-sectional study for your revised manuscript as supplementary material for peer-review only; this checklist can be easily built in https://www.goodreports.org/

Minor comments:

  1. Please review the official abbreviations of COVID-19 and SARS-CoV-2 by the WHO and modify accordingly: https://www.who.int/emergencies/diseases/novel-coronavirus-2019/technical-guidance/naming-the-coronavirus-disease-(covid-2019)-and-the-virus-that-causes-it

We thank Reviewer #1 for highlighting this, the wording of the manuscript has been updated accordingly. 

  1. Lines 75-80: This should be described in the methods section, not the introduction.

We thank Reviewer #1 for this suggestion, we have amended the introduction section accordingly.

  1. Lines 80-81: Please delete this phrase since it conveys a conclusion which should not be in the introduction section.

We thank Reviewer #1 for this suggestion, we have amended the introduction section accordingly.

  1. The specific study design has not been adequately labeled in the methods section.

We thank Reviewer #1 for this suggestion, we have amended the methods section accordingly.

  1. Inclusion and exclusion criteria should not be complementary between each other. The information that is already conveyed within an inclusion criterion does not need to be mentioned in a complementary exclusion criteria.

We thank Reviewer #1 for this suggestion, we have amended the methods section accordingly.

  1. Lines 98-100: This is part of your results, not your methods.

We thank Reviewer #1 for this suggestion, we have amended the methods section accordingly.

  1. Lines 109-110: This should be described alongside selection criteria, not here.

We thank Reviewer #1 for this suggestion, we have amended the methods section accordingly.

  1. I recommend including this reference in your discussion (DOI: 10.3390/nu13093213), because thecomponents assessed in that study, such as changes in lifestyle and weight, can affect emotional health, which were not evaluated in this study.

We thank Reviewer #1 for this suggestion, we have incorporated this referenced into the discussion section of the revised manuscript.

  1. The authors mention that they calculated 95% confidence intervals, but they are never expressed in results. Please clarify it.

We apologise, this was an error in the original manuscript and has now been removed.

Reviewer 2 Report

Manuscript: “The Psychological Impact of Coronavirus Disease 2019 (COVID-19) on its Survivors Reporting to a Tertiary Healthcare Facility in Pakistan”

The topic of the manuscript “The Psychological Impact of Coronavirus Disease 2019 (COVID-19) on its Survivors Reporting to a Tertiary Healthcare Facility in Pakistan” seems interesting for Healthcare readers. Some comments are provided to help improve the work:

Abstract: The abstract of a manuscript is a fundamental part of the study, since it is sometimes the most accessible part for readers. Authors are recommended to rewrite the abstract. Likewise, a contextualization/justification of the import of the study must be provided in the first part of the abstract. It is recommended to review the following manuscript: https://www.jclinepi.com/article/S0895-4356(13)00021-8/fulltext?s=08

Introduction:
 Authors are suggested to use part of the introduction to justify the importance of knowing the mental health situation of the population. The implications that affected mental health has at the public health level must be made clear.
 In the introduction I do not think it is necessary to include examples of tools to measure the main study variables. I believe that this is a piece of information that should appear specifically in the methodology or material and methods section.
 The last paragraph of the introduction (lines 73-81) presents data that should appear in the methodology or material and methods. Instead, the introduction should end with a clear and concise objective(s) of the study.
 It is recommended to look at the following document of interest: https://www.jclinepi.com/article/S0895-4356(13)00020-6/fulltext?s=08

Materials and methods:
 The authors must include in this section a specific section on "study design" or similar in which they include the type of study they are proposing, as well as interesting information for the reader, such as whether there is a previously published study protocol.
 Was the sample selected a convenience sample? Why were patients recruited at that particular hospital? How is the sample size calculated? All of these data should be included in the methodology, in order to make the study transparent and advocate for reproducible science.
 I understand that the sociodemographic data collected was collected through an ad hoc questionnaire, but this should appear as well as the type of questions used (open, closed with specific response options...). Collecting the "state of the patient" is very ambiguous; all this should be specified in this section.
 Are the scales used (DASS-21 and PCL-5) validated in the Pakistani population? Cultural variation can influence the experience and emotional expression of the participants invalidating the results of any study. They should include information on the internal consistency of these scales in this study.
 At the methodological level, an association cannot be determined if a regression analysis has not been performed. This is completely wrong and the authors must correct it. Likewise, the statistical analysis section must be completed including missing information such as, has the normality of the quantitative variables been verified? Likewise, all measures of central tendency must be accompanied by their measure of dispersion, this must appear literally in this section and must be applied in the results.

Results:
 Authors should reflect on the values ​​they show. P-values ​​are shown directly without showing any other measures. What does this mean? Association is discussed and no measure of association is shown.

Discussion:
 In the first paragraph, the authors should only show the results of their study. No result should be discussed that has not been mentioned in this first paragraph.
 The results must be interpreted considering "pros" and "cons" and, taking into account other similar investigations (if any) and provide a plausible explanation of their results.
 All limitations of the study should be acknowledged, addressing sources of potential bias, imprecision…
 The authors should discuss the generalizability of their results based on external and internal validity.

Conclusion:
 The conclusion does not seem to respond to the study hypothesis and should be reworked as well as synthesized.

Author Response

Manuscript: “The Psychological Impact of Coronavirus Disease 2019 (COVID-19) on its Survivors Reporting to a Tertiary Healthcare Facility in Pakistan”

The topic of the manuscript “The Psychological Impact of Coronavirus Disease 2019 (COVID-19) on its Survivors Reporting to a Tertiary Healthcare Facility in Pakistan” seems interesting for Healthcare readers. Some comments are provided to help improve the work:

Abstract: The abstract of a manuscript is a fundamental part of the study, since it is sometimes the most accessible part for readers. Authors are recommended to rewrite the abstract. Likewise, a contextualization/justification of the import of the study must be provided in the first part of the abstract. It is recommended to review the following manuscript: https://www.jclinepi.com/article/S0895-4356(13)00021-8/fulltext?s=08

We thank Reviewer #2 for the suggestion of considering the abstract in more depth and adjusting the scientific writing. A clear need for the study should have been stated in the abstract and likewise how that need was going to be addressed.  These have now been added to the abstract of the revised manuscript and other wording has also been amended in response to the recommendation of Reviewer #3, and the need to trim the abstract length to approximately 200 words, in keeping with the journal guidelines.

Introduction:
§ Authors are suggested to use part of the introduction to justify the importance of knowing the mental health situation of the population. The implications that affected mental health has at the public health level must be made clear.

We appreciate the need to highlight the importance of healthcare support for individuals with mental health conditions.  We have therefore added text to the introduction that precedes the justification for the study.

  • In the introduction I do not think it is necessary to include examples of tools to measure the main study variables. I believe that this is a piece of information that should appear specifically in the methodology or material and methods section.

We appreciate the point made by Reviewer #2 that there is a need to limit methodology to the materials and methods section.  We included references to other related studies in the introduction section to highlight the methods that have been employed to assess the mental health of the study population.  However, we concur that some of the details concerning the questionnaire used for our study could be covered in the materials and methods section.  Therefore, this text has been moved from the introduction to the materials and methods section in the revised manuscript.

  • The last paragraph of the introduction (lines 73-81) presents data that should appear in the methodology or material and methods. Instead, the introduction should end with a clear and concise objective(s) of the study.
    § It is recommended to look at the following document of interest: https://www.jclinepi.com/article/S0895-4356(13)00020-6/fulltext?s=08

We thank Reviewer #2 for this suggestion. We amended the introduction to remove the text covering methodology, and have included a clear objective for the study.

Materials and methods:
§ The authors must include in this section a specific section on "study design" or similar in which they include the type of study they are proposing, as well as interesting information for the reader, such as whether there is a previously published study protocol.

We thank reviewer #2 for the suggestion of more detail about the study design and this has been included in the revised document.  The study was a cross-sectional one with patients recruited using convenience sampling.

  • Was the sample selected a convenience sample? Why were patients recruited at that particular hospital? How is the sample size calculated? All of these data should be included in the methodology, in order to make the study transparent and advocate for reproducible science.

We thank Reviewer #2 for these suggestions to improve the materials and methods section.  We can confirm that the patient participants were recruited via convenience sampling and that this hospital was used for the study because it is one of the largest healthcare facilities in Pakistan that was dealing with COVID-19 patients.  Sample size related to a manageable number of patients These details have been added to the revised manuscript.

  • I understand that the sociodemographic data collected was collected through an ad hoc questionnaire, but this should appear as well as the type of questions used (open, closed with specific response options...). Collecting the "state of the patient" is very ambiguous; all this should be specified in this section.

The sociodemographic data could be better presented and has now been incorporated into each of the Tables 1-4 covering depression, anxiety, stress and PTSD. We do provide a link to the questionnaire for the patients (Supplementary Section) which is mainly comprised of closed questions with specific response options. The current condition of the patient was compared with their health condition prior to the disease.  In this demographic data section, we now make a further reference (in the revised manuscript) to the study questionnaire within the Supplementary Section.

  • Are the scales used (DASS-21 and PCL-5) validated in the Pakistani population? Cultural variation can influence the experience and emotional expression of the participants invalidating the results of any study. They should include information on the internal consistency of these scales in this study.

Reviewer #2 makes the important point about study validation in the Pakistani population. There have been some studies that have considered screening tools for validation in the Pakistani population (Mumford et al. 2005 Development and validation of a questionnaire for anxiety and depression in Pakistan. J Affect Disord. 2005 Oct;88(2):175-82) and Karmaliani et al., 2007 Diagnostic validity of two instruments for assessing anxiety and depression among pregnant women in Hyderabad, Pakistan. Health Care Women Int. 2007 Jul;28(6):556-72).  More recently, a review was undertaken that considered the use of screening tools in middle-income countries (Mughal et al 2020 A systematic review of validated screening tools for anxiety disorders and PTSD in low to middle income countries. BMC Psychiatry. 2020;20(1):338). Collectively, the DASS-21 scale has been used in both local as well as regional studies in assessing the mental health of the population in Pakistan (reference 9 of the manuscript), and has been used by others when considering the impact of COVID-19, including those of Bangladesh (references 16-18 of the original manuscript) so was deemed a useful and viable means of assessment.  The use of PCL-5 has recently been validated in Bangla (Islam et al. 2020 The psychometric properties of the Bangla Posttraumatic Stress Disorder Checklist for DSM-5 (PCL-5): preliminary reports from a large-scale validation study. BMC Psychiatry 22, 280), and has been used in other countries such as China to assess PTSD in COVID-19 survivors (Tu et al 2021 Post-traumatic stress symptoms in COVID-19 survivors: a self-report and brain imaging follow-up study. Mol Psychiatry 26, 7475–7480 (2021).  We have therefore provided details of these papers and into the methods section of the revised manuscript.

  • At the methodological level, an association cannot be determined if a regression analysis has not been performed. This is completely wrong and the authors must correct it. Likewise, the statistical analysis section must be completed including missing information such as, has the normality of the quantitative variables been verified? Likewise, all measures of central tendency must be accompanied by their measure of dispersion, this must appear literally in this section and must be applied in the results.

We have amended the methodological details and associated statistics with a Chi-squared test performed for a linear trend.

Results:
§ Authors should reflect on the values ​​they show. P-values ​​are shown directly without showing any other measures. What does this mean? Association is discussed and no measure of association is shown.

We apologise for providing insufficient statistical data and validation.  The data tables and the methods were updated to state that a Chi-squared test was performed for a linear trend. The p-value calculated (and displayed in the Tables) was used to determine if there was a relationship between the categorical variables. 

Discussion:
§ In the first paragraph, the authors should only show the results of their study. No result should be discussed that has not been mentioned in this first paragraph.

The first paragraph of the discussion has been reworked in the revised manuscript to only cover the results of the study, as recommended by Reviewer #2.

  • The results must be interpreted considering "pros" and "cons" and, taking into account other similar investigations (if any) and provide a plausible explanation of their results.

The results have been considered further and more context is provided by considering the pros and cons of the results as well as their relevance to other published studies.

  • All limitations of the study should be acknowledged, addressing sources of potential bias, imprecision…

We appreciate the need to consider the limitations of the study and have addressed this more fully in the discussion of the revised manuscript.

  • The authors should discuss the generalizability of their results based on external and internal validity.

We thank reviewer #2 for this suggestion and have added more commentary to the discussion that covers the generalizability of the results.

Conclusion:
§ The conclusion does not seem to respond to the study hypothesis and should be reworked as well as synthesized.

The concluding statements have been reworded to better align with the study hypothesis.

Reviewer 3 Report

The paper is in general need of wordsmithing and several areas of the manuscript must be further substantiated or clarified.

Specific comments:

  1. Please shorten the abstract to approximately 200 words as per the journal's guidelines.
  2. Please change "Corona Virus Disease-2019 (COVID-19)" to "Coronavirus Disease 2019 (COVID-19)".
  3. Please omit "novel coronavirus (2019-nCoV)". This is an old terminology that is no longer in use.
  4. Instead of ref [10], suggest to cite a more recent review on the topic (citation: pubmed.ncbi.nlm.nih.gov/32603985).
  5. "... 13.5% and 7.7% were of the severe and extremely severe categories" - when presenting results, please present the actual n and its accompanying percentage.
  6. Please change "chronic psy-chological" to "chronic psychological" and "re-spectively" to "respectively".
  7. The 95% confidence intervals should also be reported in Table 4. It is difficult to interpret p values in isolation.
  8. "... post-traumatic psychological effects of COVID-19 on its survivors" - the manuscript should drop the word 'survivor' in favour of 'recovered' instead, especially since majority of the patients recruited had only mild to moderate symptoms. Given that a significant proportion of the world's population is now COVID-recovered, recovered would be a more apt choice of word than survived.
  9. "These results are comparable to the study of Tan et al. (2020)[10] which also reported higher levels of anxiety (20.7%) among non-medical healthcare workers when compared with medical personnel (10.8%) in Singapore" - this is not comparable as the study did not look at a COVID-recovered population and was performed relatively early in the pandemic, when information and knowledge about COVID-19 was particularly lacking.
  10. As resources could be particularly scarce during a serious pandemic situation, timely psychological support could also take many forms, including telemedicine and informal support groups (citation: pubmed.ncbi.nlm.nih.gov/32380875). This should be mentioned.
  11. "... damaged the world’s population physically" - very awkward phrasing. Please rephrase.
  12. Citation style inconsistent.

Author Response

Reviewer #3, specific comments:

  1. Please shorten the abstract to approximately 200 words as per the journal's guidelines.

We thank Reviewer #3 for the prompt on this and have reduced the word count to approximately 200 words as requested.  The text has been reworded and parts deleted.

  1. Please change "Corona Virus Disease-2019 (COVID-19)" to "Coronavirus Disease 2019 (COVID-19)".

We thank Reviewer #3 for the suggestion to update the terminology and this has been corrected in the revised manuscript.

  1. Please omit "novel coronavirus (2019-nCoV)". This is an old terminology that is no longer in use.

We thank Reviewer #3 for this suggestion and amended the wording of the revised manuscript.

  1. Instead of ref [10], suggest to cite a more recent review on the topic (citation: pubmed.ncbi.nlm.nih.gov/32603985).

We thank Reviewer #3 for this suggestion.  The paper suggested is a narrative review of observational studies on the mental health effects of COVID-10 on healthcare workers and therefore includes a number of relevant studies that specifically focus on this issue.  We concur that it would be a useful reference for the reader that covers more fully the impact of COVID-19 specifically on healthcare workers rather than reference [10].  As suggested, we have therefore replaced reference [10] with the suggested reference (Ng et al., 2020).

  1. "... 13.5% and 7.7% were of the severe and extremely severe categories" - when presenting results, please present the actual n and its accompanying percentage.

We appreciate that within the text of the results section we could also include this information regarding the numbers of study participants that experienced depression, as well as listing these in the Tables.  We have therefore now included this information in the text of the document as well as the numbers that experienced anxiety and stress.

  1. Please change "chronic psy-chological" to "chronic psychological" and "re-spectively" to "respectively".

These are not spelling/typographical errors in the original document but are generated in the text at these (and other points) by the journal formatting.

  1. The 95% confidence intervals should also be reported in Table 4. It is difficult to interpret p values in isolation.

We appreciate that the presentation of the data could be improved and therefore new Tables (1-4) have been generated that provide all of the study details, results and statistical analysis for depression, anxiety, stress and PTSD.

  1. "... post-traumatic psychological effects of COVID-19 on its survivors" - the manuscript should drop the word 'survivor' in favour of 'recovered' instead, especially since the majority of the patients recruited had only mild to moderate symptoms. Given that a significant proportion of the world's population is now COVID-recovered, recovered would be a more apt choice of word than survived.

We agree with Reviewer #3, that recovery or recovered from, is more in keeping with the current situation with COVID-19.  We have therefore amended this wording in the revised manuscript.

  1. "These results are comparable to the study of Tan et al. (2020)[10] which also reported higher levels of anxiety (20.7%) among non-medical healthcare workers when compared with medical personnel (10.8%) in Singapore" - this is not comparable as the study did not look at a COVID-recovered population and was performed relatively early in the pandemic when information and knowledge about COVID-19 were particularly lacking.

We appreciate the point made by Reviewer #3, the word comparable relates to the numbers of individuals affected that are similar but does not consider the study design, cohort size, etc.  We have therefore amended this wording to remove comparable and simply highlighted other studies, such as the reference cited, which have also recorded higher levels of anxiety in non-medical healthcare workers when compared with medical personnel.

  1. As resources could be particularly scarce during a serious pandemic situation, timely psychological support could also take many forms, including telemedicine and informal support groups (citation: pubmed.ncbi.nlm.nih.gov/32380875). This should be mentioned.

We agree with Reviewer #3 that adaptation of current healthcare practices will assist with the treatment and support required for COVID-19 patients.   We have therefore included additional details to the discussion section that covers this and cites the suggested reference.

  1. "... damaged the world’s population physically" - very awkward phrasing. Please rephrase.

We concur with Reviewer #3 and have amended the wording of the conclusion section.

  1. Citation style inconsistent.

The citation style used throughout is Vancouver in keeping with the journal requirements.  We have, on occasion, cited work directly as well (author et al and year in brackets) but we have also included the reference in brackets in keeping with the Vancouver style.

Round 2

Reviewer 1 Report

In the corrected version of this manuscript, the authors have addressed some of my prior comments, however most of my prior comments were only partially answered on the responses documents, but the solicited changes and clarifications are not reflected in the manuscript. I am currently not able to fairly assess if this manuscript may be suitable for publication or not since the authors decided not to adhere to STROBE recommendations and did not provide the STROBE checklist as supplementary material. I will not be able to endorse publication of this manuscript if the authors continue to avoid adhering to reporting standards. https://journals.plos.org/plosmedicine/article?id=10.1371/journal.pmed.0040297

Comments:

1. The objective of the study was changed, but the objective in the abstract is not clearly defined. I suggest that the authors place the same objective of the study written in the body of the manuscript in the abstract.

2. The authors made modifications in their manuscript about the limitations of their study, but these were not placed in the discussion, but in the conclusions, this is an error, and the text should be modified. The limitations of line 279 to 299 should go into the discussion section.

3. The conclusion section should be a single paragraph. The current summary and conclusions section is inadequate. Please move most of these ideas to the discussion section. 

4. The current conclusion in the abstract is inadequate since a cross-sectional study cannot be used to conclude if patients will benefit from receiving treatment or support: “Collectively, these results indicate that patients recovering from COVID-19 experience psychological distress that will require ongoing treatment and support.” Please change your conclusions to reflect only your main findings. 

5. Lines 96-97 is part of the results and should not be in materials and methods.

6. Definition of exposures and outcomes was requested in the previous review, the authors mention that they were defined, but they are nowhere to be found in the materials and methods of the new manuscript.  This is very important to be able to assess if this manuscript could be suitable for publication or not. Furthermore, presentation of results will need to be done according to these variables. 

7. I again insist that the authors include a flowchart according to STROBE recommendations.

8. The question in the previous version about the severity of the disease was partially answered, but the question remains on how was it possible to know all the information that the authors mentioned since the methods explain that the survey was developed online and applied only when the patients went to a follow-up consultation at the hospital.

9. The classification placed in materials and methods on the DASS 21 is not defined by the scores. Each category must be defined according to the scores obtained.

10. The authors did not include the requested STROBE checklist in the version of the review, again I ask the authors to include it so that it can be verified that each reporting point requested by STROBE was met.

11. The authors were asked to correctly define the type of study they carried out in the methods section, which was not clarified in the manuscript. Please attend all requests.

12. The title does not mention the study design according to STROBE recommendations. 

13. The authors need to review the correct abbreviations for COVID-19 and SARS-CoV-2 according to the World health Organization to correct the entire manuscript accordingly: https://www.who.int/emergencies/diseases/novel-coronavirus-2019/technical-guidance/naming-the-coronavirus-disease-(covid-2019)-and-the-virus-that-causes-it

Author Response

In the corrected version of this manuscript, the authors have addressed some of my prior comments, however most of my prior comments were only partially answered on the responses documents, but the solicited changes and clarifications are not reflected in the manuscript. I am currently not able to fairly assess if this manuscript may be suitable for publication or not since the authors decided not to adhere to STROBE recommendations and did not provide the STROBE checklist as supplementary material. I will not be able to endorse publication of this manuscript if the authors continue to avoid adhering to reporting standards. https://journals.plos.org/plosmedicine/article?id=10.1371/journal.pmed.0040297

  • We thank reviewer #1 for detailed and critical analysis. We have made changes according to STROBE checklist. Details of the checklist and actions taken are as follows:

STROBE Items Checklist

Actions Taken

TITLE and ABSTRACT:

1. (a) Indicate the study’s design with a commonly used term in the title or the abstract

    (b) Provide in the abstract an informative and balanced summary of what was done and what was found

Study design indicated in the title as well as abstract

Informative and Balanced summary provided in the abstract

INTRODUCTION:

Background/rationale

2. Explain the scientific background and rationale for the investigation being reported

Objectives

3. State specific objectives, including any pre-specified hypotheses

Scientific background with references mentioned in paragraphs 1-4 of the introduction

Specific objectives of the study detailed in paragraph 5 of the introduction

METHODS:

Study design

4. Present key elements of study design early in the paper

Setting

5. Describe the setting, locations, and relevant dates, including periods of recruitment, exposure, follow-up, and data collection

Participants

6. Cross-sectional study—Give the eligibility criteria, and the sources and methods of selection of participants

Variables

7. Clearly define all outcomes, exposures, predictors, potential confounders, and effect modifiers. Give diagnostic criteria, if applicable

Data sources/ measurement

8. For each variable of interest, give sources of data and details of methods of assessment (measurement). Describe comparability of assessment methods if there is more than one group

Bias

9. Describe any efforts to address potential sources of bias

Study size

10. Explain how the study size was arrived at

Quantitative variables

11. Explain how quantitative variables were handled in the analyses. If applicable, describe which groupings were chosen, and why

Statistical methods

12. (a) Describe all statistical methods, including those used to control for confounding

      (b) Describe any methods used to examine subgroups and interactions

      (c) Explain how missing data were addressed

      (d) Cross-sectional study—If applicable, describe analytical methods taking account of sampling strategy

      (e) Describe any sensitivity analyses

Study design and setting listed in the first paragraph of the methods.

Settings, locations, relevant dates and exposure mentioned in the first paragraph of the methods.

Eligibility criteria and method of selection covered in the second paragraph of the methods section.

Outcomes, exposures, predictors and effect modifiers (where relevant) defined.

Information regarding source of data and methods of assessment (i.e. DASS-21 scale and PCL-5) provided.

Patients were asked to answer structured questionnaire so that interviewer bias could be eliminated.

Study size measurements have been detailed.

Explanation regarding the quantitative variables have been included.

The statistical method used is detailed.

Sub-group examination method is detailed.

All data was analyzed.

The sampling method was detailed.

Not applicable.

RESULTS:

Participants

13. (a) Report the numbers of individuals at each stage of the study—e.g., numbers potentially eligible, examined for eligibility, confirmed eligible, included in the study, completing follow-up, and analyzed

      (b) Give reasons for non-participation at each stage

      (c) Consider use of a flow diagram

Descriptive data

14. (a) Give characteristics of study participants (e.g., demographic, clinical, social) and information on exposures and potential confounders

      (b) Indicate the number of participants with missing data for each variable of interest

Outcome data

15. Cross-sectional study—Report numbers of outcome events or summary measures

Main results

16. (a) Give unadjusted estimates and, if applicable, confounder-adjusted estimates and their precision (e.g., 95% confidence interval). Make clear which confounders were adjusted for and why they were included

      (b) Report category boundaries when continuous variables were categorized

      (c) If relevant, consider translating estimates of relative risk into absolute risk for a meaningful time period

Other analyses

17. Report other analyses done—e.g., analyses of subgroups and interactions, and sensitivity analyses

Details of the number of eligible individuals have been included in the study.

It was a single stage study (cross-sectional) study design.

A flow diagram has been added.

Demographics and clinical information of the study participants has been included.

Number of participants not completing the study have been included, missing data not applicable.

Number of outcome events detailed (i.e. proportion of participants that experienced depression, anxiety, depression, stress and PTSD have been included).

Not applicable.

Categories of variables clearly defined (i.e. normal to extremely severe)

Not applicable

Detailed analysis of groups and subgroups are provided.

DISCUSSION:

Key results

18. Summarize key results with reference to study objectives

Limitations

19. Discuss limitations of the study, taking into account sources of potential bias or imprecision. Discuss both direction and magnitude of any potential bias

Interpretation

20. Give a cautious overall interpretation of results considering objectives, limitations, multiplicity of analyses, results from similar studies, and other relevant evidence

Generalizability

21. Discuss the generalizability (external validity) of the study results

Key results in line with the study objectives are summarized in the discussion section.

Study limitations are detailed.

Overall interpretation of the study results has been included.

Generalizability of the study has been discussed.

OTHER INFORMATION:

Funding

22. Give the source of funding and the role of the funders for the present study and, if applicable, for the original study on which the present article is based

There was no external funding for the study and this has been mentioned.

Comments:

  1. The objective of the study was changed, but the objective in the abstract is not clearly defined. I suggest that the authors place the same objective of the study written in the body of the manuscript in the abstract.

Ans: The objective of the study is clear within the abstract in keeping with the body of the manuscript.

  1. The authors made modifications in their manuscript about the limitations of their study, but these were not placed in the discussion, but in the conclusions, this is an error, and the text should be modified. The limitations of line 279 to 299 should go into the discussion section.

Ans: The study limitations have been moved to the discussion section.

  1. The conclusion section should be a single paragraph. The current summary and conclusions section is inadequate. Please move most of these ideas to the discussion section. 

Ans: The summary and conclusion section has been changed to a single paragraph.

  1. The current conclusion in the abstract is inadequate since a cross-sectional study cannot be used to conclude if patients will benefit from receiving treatment or support: “Collectively, these results indicate that patients recovering from COVID-19 experience psychological distress that will require ongoing treatment and support.” Please change your conclusions to reflect only your main findings. 

Ans: The concluding statement of the abstract has been changed.

  1. Lines 96-97 is part of the results and should not be in materials and methods.

Ans: Lines 96-97 has been moved to the results section.

  1. Definition of exposures and outcomes was requested in the previous review, the authors mention that they were defined, but they are nowhere to be found in the materials and methods of the new manuscript.  This is very important to be able to assess if this manuscript could be suitable for publication or not. Furthermore, presentation of results will need to be done according to these variables.

Ans: Definitions of exposures have been added to the revised manuscript.

  1. I again insist that the authors include a flowchart according to STROBE recommendations.

Ans: A flowchart as per STROBE recommendations has been added.

  1. The question in the previous version about the severity of the disease was partially answered, but the question remains on how was it possible to know all the information that the authors mentioned since the methods explain that the survey was developed online and applied only when the patients went to a follow-up consultation at the hospital.

Ans: The survey was developed online as well as on paper and more than 90% of the data was collected through paper responses rather than online. Additionally, details regarding the requirement of a medical record detailing the severity of a patient’s COVID-19 illness have been added.

  1. The classification placed in materials and methods on the DASS 21 is not defined by the scores. Each category must be defined according to the scores obtained.

Ans: Classification of categories with scores have been added as per DASS-21.

  1. The authors did not include the requested STROBE checklist in the version of the review, again I ask the authors to include it so that it can be verified that each reporting point requested by STROBE was met.

Ans: The STROBE checklist of recommendations for observational studies have been added above.

  1. The authors were asked to correctly define the type of study they carried out in the methods section, which was not clarified in the manuscript. Please attend all requests.

Ans: The type of study has been mentioned in the title, abstract as well as in the study design.

  1. The title does not mention the study design according to STROBE recommendations

Ans: The title of the study has been changed as per STROBE guidelines. 

  1. The authors need to review the correct abbreviations for COVID-19 and SARS-CoV-2 according to the World health Organization to correct the entire manuscript accordingly: https://www.who.int/emergencies/diseases/novel-coronavirus-2019/technical-guidance/naming-the-coronavirus-disease-(covid-2019)-and-the-virus-that-causes-it

Ans: Abbreviations of COVID-19 and SARS-CoV-2 have been updated as per WHO guidance.

Reviewer 3 Report

Thank you for the revisions.

Specific comments:

1. Please change "patients reporting to" to "patients admitted to".

2. The study title and objectives remain befuddled and unclear. I suggest authors reword them.

Author Response

Thank you for the revisions.

Specific comments:

  1. Please change "patients reporting to" to "patients admitted to".

Thank you Reviewer #3 for this suggestion, we have amended the text in the revised manuscript to "patients admitted to".

2. The study title and objectives remain befuddled and unclear. I suggest authors reword them.

Thank you Reviewer #3 for this suggestion, we have amended the text of the title, abstract, and other sections of the manuscript to make the objectives of the study clear.